# Leadership and governance for integrating mental healthcare at the primary healthcare (PHC) level: A mixed methods study in Ghana

Peter Badimak Yaro[1¤a☯]*, Emmanuel Asampong[2☯], Philip Teg-Nefaah Tabong[2‡], Graham Thornicroft[3‡], Paulina Tindana[1☯]

1 Department of Health Policy, Planning and Management, School of Public Health, Box LG 13, College of Health Sciences, University of Ghana, Legon, Accra, Ghana, 2 Department of Social and Behavioural sciences, School of Public Health, BOX LG 13, College of Health Sciences, University of Ghana, Legon, Accra, Ghana, 3 Health Service and Population Research Department, Institute of Psychiatry, King's College London, London, United Kingdom

☯ These authors contributed equally to this work.
‡ These authors also contributed equally to this work.
¤a Current Address: BasicNeeds-Ghana, Box TL 1140, Tamale, Ghana
* peter.yaro@basicneedsghana.org, bpyaro@st.ug.edu.gh

## Abstract

Leadership and governance are key components of health systems, nevertheless research into leadership and governance of mental healthcare at the Primary Health Care (PHC) level is probably the least well researched and understood part of these systems. As part of assessing the integration of mental health at the PHC level in Ghana, the leadership and governance organisation and structures to ensure supervision and coordination were examined. A concurrent triangulation mixed-methods design involving both quantitative and qualitative research methods approach was adopted. The quantitative data were collected through a questionnaire, self-administered or interviewer administered, on 1010 respondents with 830 completed (response rate 82%). Key informant interviews and focus group discussions were used to collect the qualitative data. Thematic content analysis utilising NVivo 12 was applied for the qualitative field data. Stata SE16 was used for quantitative data. Data triangulation strategy was used to report the qualitative and quantitative results. The study showed that leadership and governance of mental health at the PHC level were partially developed, with a composite mean score of 2.53, due to the modest level of awareness of the Mental Health Law, inadequate functioning and coordination of mental health units, low private sector participation in mental health care services, and low levels of monitoring, supervision, and evaluation. This affected the integration of mental health at the PHC level, which was also gauged as low. The study concludes that despite the presence of legislation and policy aiming to achieve decentralised and integrated mental health services at the PHC level,

**Data availability statement:** The underlying data is contained in the paper and its Supporting Information files. Data is also available on request to the corresponding author (peter.yaro@basicneedsghana) or the Secretary to the Ghana Health Service Ethics Review Committee (ethics.research@ghs.gov.gh).

**Funding:** The author(s) received no specific funding for this work.

**Competing interests:** The authors have declared that no competing interests exist.

mental health care is still a low-level priority within the health care system in Ghana and tends to operate within a silo. The study recommends that more practical and concerted leadership of mental health at the regional and district levels is required to drive decentralisation and integration at these levels.

## Introduction

Only a small proportion of people living with mental health conditions receive treatment, despite effective interventions that address these conditions, prevent associated impact, reduce their prevalence, and promote overall mental wellbeing overall [1–5]. The scale-up of mental health care and people-centred community-based mental health care service models are best achieved through integration into general healthcare at the Primary Health Care (PHC) level [4,6–9]. Integration of mental healthcare into general healthcare at the Primary Health Care (PHC) level remains the core and singular global consensus to scaling-up mental healthcare and addressing the high treatment gap in resource limited settings, especially in Low- and Middle-Income Countries (LMICs) [10–13].

Integration of mental health care within the general health care system in PHC is best appreciated within the context of the interaction of the components of the Health System Framework proposed by the WHO and further enhanced by the Ouagadougou Declaration on Primary Health Care and Health Systems in Africa [14,15]. The WHO Health System Framework is bringing together an interaction of the various building blocks that constitute a health system to be able to deliver health care services to meet the health needs of the population [16]. The key building blocks of the health system include: (i) Leadership and Governance; (ii) health service delivery; (iii) health workforce/ health human resources; (iv) health information systems; (v) health products, vaccines and technologies; (vi) health financing; (vii) community ownership and participation; (viii) partnership for health development; and (ix) research for health [17].

In recent years, the growth in understanding and appreciation of the importance of mental health in population health and wellbeing have heightened the need to develop mental health systems to respond to the mental health care service needs of populations [13,18]. Leadership and governance are important components of the health system framework in the development and integration of mental health policy and services at the community level. They are considered the hub around which the various components of the health system framework revolve [17]. The WHO succinctly sets out that leadership and governance involves ensuring the existence of a strategic policy framework which combines "effective oversight [supervision, monitoring and evaluation], coalition-building, regulation and attention to system-design and accountability" [19]. It takes the interactions of three categories of stakeholders to determine the health system and its leadership andgovernance [20–22]. These include, the state, the healthcare service providers, and the citizens, particularly the healthcare service users. 'State' covers the government Ministries, Departments and Agencies (MDAs) at the central and regional and local levels. 'Healthcare service providers' refer to public

and private for profit and not-for-profit clinical providers, para-medical and non-medical healthcare providers, unions, and medical/ health professional associations, as well as networks of care or of services. The third category of stakeholders in the health system leadership and governance is the 'citizens', particularly population representatives such as user/patient associations, and Non-Governmental Organisations (NGOs) and Civil Society Organisations (CSOs).

Essential to effective leadership and governance for integrated mental health system are legislation, policies, and strategies for ensuring coordination and oversight [23–25]. Ghana's mental health law, mental health policy, and strategic plan affirm the global understanding of what a health system thinking is [26–28]. Key provisions of the Mental Health Law (Act 846; 2012) and the overall purpose and interpretation of the Legislation Instrument emphasise decentralisation of mental healthcare services down to the community level, integrated into general healthcare services at the PHC level. The expected decentralisation and integration of mental health care into general care at the community level in Ghana, remains rather limited in practice and far less successful overall [29,30].

Mental health care service provision in Ghana remains largely a silo in the healthcare delivery arrangements of the country, especially at the community level, perpetuating the disconnection, isolation, and discrimination of mental health in general health care services across the country [31]. The bulk of resources for mental health care (financial, human and logistical/ medicines) go to the three psychiatric hospitals of Ghana, located in the southern-coastal regions, rather than the regional, district and lower levels of healthcare delivery which would serve a bigger population than the three psychiatric hospitals do. This situation has contributed to the low resourcing and investments in mental health at the lower levels of the healthcare system, and a generally low priority accorded mental health.

A most likely problem of such situation with mental health in Ghana could be leadership and governance inadequacies. Since the passage of the Mental Health Act (Act 846), in 2012, there has not been a critical examination and clarification of the roles and responsibilities of the Mental Health Authority (MHA) and the Ghana Health Service (GHS) in the coordination/ management structure, organisation, and delivery of mental health care services outside of the psychiatric hospitals, for that matter at the PHC level. The MHA draws heavily from the Mental Health Act (846) to operate while the GHS substantially relies on the Public Service Act, 1996 (525) to deliver its mandate as the agency that administers health services of Ghana and implements government policies on healthcare [32,33]. These two legally backed roles have somewhat affected the way and manner community mental health services are organised, managed, and delivered. The Mental Health Act, 2012 (846) was promulgated to address the years of neglect of the sub-sectors affecting the organisation of services and overall oversight and leadership in the country [34]. A decade following the promulgation of this Act in Ghana and nearly eight years of the coming into being of the MHA operating along with the GHS, the intended structure and organisation of mental health services outside of the psychiatric hospitals have remained blurred. Improving and sustaining clear leadership and governance of community mental health is integral to overall health system strengthening efforts.

There is, therefore, the interest and need to examine the key dynamics and undertakings that could promote integration of mental healthcare at the PHC level of the Ghana's healthcare system. The aim of this study was therefore to assess the leadership and governance in place for the integration of mental healthcare at the PHC in Ghana. This is part of understanding the supervision, monitoring and coordination and general stewardship of mental healthcare integration in Ghana.

## Materials and methods

### Ethics statement

Ethical, cultural, and scientific considerations specific to inclusivity to global research were adhered to. Research ethics approval was obtained from the GHS Ethics Review Committee (ERC) with the approval letter numbered GHS-ERC018/12/19. Formal letters of permission and introduction(s) were obtained from the various directors and in-charges of regions, districts, and healthcare facilities, as well as verbal approvals from community authorities and clan/family heads. Informed consent was obtained from the study participants using the GHS-ERC approved research study information brochure and informed consent form.

## Study design

This study adopted a concurrent triangulation mixed-methods design [35–39]. It involved the use of both qualitative and quantitative approaches [40–42]. The primary purpose for the qualitative and quantitative approaches is to maximise the effect of both methods in proffering a better understanding of the phenomenon being studied than will be the case with just a singular use of one of either approaches, hence minimising the bias and weaknesses of qualitative or quantitative research. A cross-sectional descriptive study design was used for the quantitative component of the study [43–45]. The qualitative method adopted a critical descriptive ethnography approach [46–48]. The aim of qualitative research data component was to generate in-depth information from the study respondents that answer the questions 'what', 'why' and 'how' of phenomena, events and practices with the aim of exploring human behaviour, attitudes and interpretation and meaning made out of their experiences and interaction with their environment [49].

## Study area

The study was conducted in the Republic of Ghana. Ghana is divided into sixteen regions (Fig 1) and three ecological zones (Northern, Middle, and Southern).

Health service delivery in Ghana is a five-tiered service system, including: National, Regional, District, Sub-district, and the Community [50–53]. The district and community level health facilities provide primary healthcare services and the secondary healthcare facilities serve as higher referral points from the primary healthcare service facilities [51,54]. In 1999, Ghana adopted the Community-based Health Planning and Services (CHPS) strategy as a primary health care system [55–58]. In the CHPS strategy, Community Health Officers (CHOs) are assigned to a demarcated CHPS zone(s) as a measure to ensure the close-to-client model of community health services in those demarcated zones [59,60].

## Study population

The study population included health policy officials, healthcare managers, healthcare service providers, persons with mental health conditions and primary caregivers of people with mental health conditions, civil society organizations, self-advocates, and mental health champions.

## Sampling and sample size determination

Stratified sampling technique was used for the quantitative component of the study [43,61]. This was so that the respondents are selected because they have characteristics needed and are suitable for the study's objectives [62,63]. It made it possible for the study to reach respondents likely, willing and ready to furnish useful information within the limited time space and at reasonable cost (labour and money). The stratification of the sample ensured adequate representation of the three categories of respondents in the study and maximised the study's efficiency and validity [63]. In this regard, the total sample for the survey was allocated equally to each of the three categories of respondents.

It was recognised that most of the national level policy officials will be from Accra, the southern-coastal zone. The use of purposive sampling also helped to address the challenge where a randomised sample would have generated subjects drawn for the study to include persons living with mental health conditions with limited or diminished capacity to give free consent and not stabilised enough to respond to the survey questionnaire, or who may not be available in their usual domiciles or workplaces during the data collection exercise. The purposive sample equally assured that service respondents with lived experience and articulate enough were reached to serve as respondents.

A sample of 1104 potential respondents for the survey was established. The sample of 0.00115403 proportion of the population of each of the eight districts as per data of the Ghana Statistical Service (GSS). The sample was arrived at based on using the Epi Info™ programme developed by the United States of America Centers for Disease Control and Prevention (CDC) [64], guided by the mental health prevalence rate and parameters for calculating sample size. The

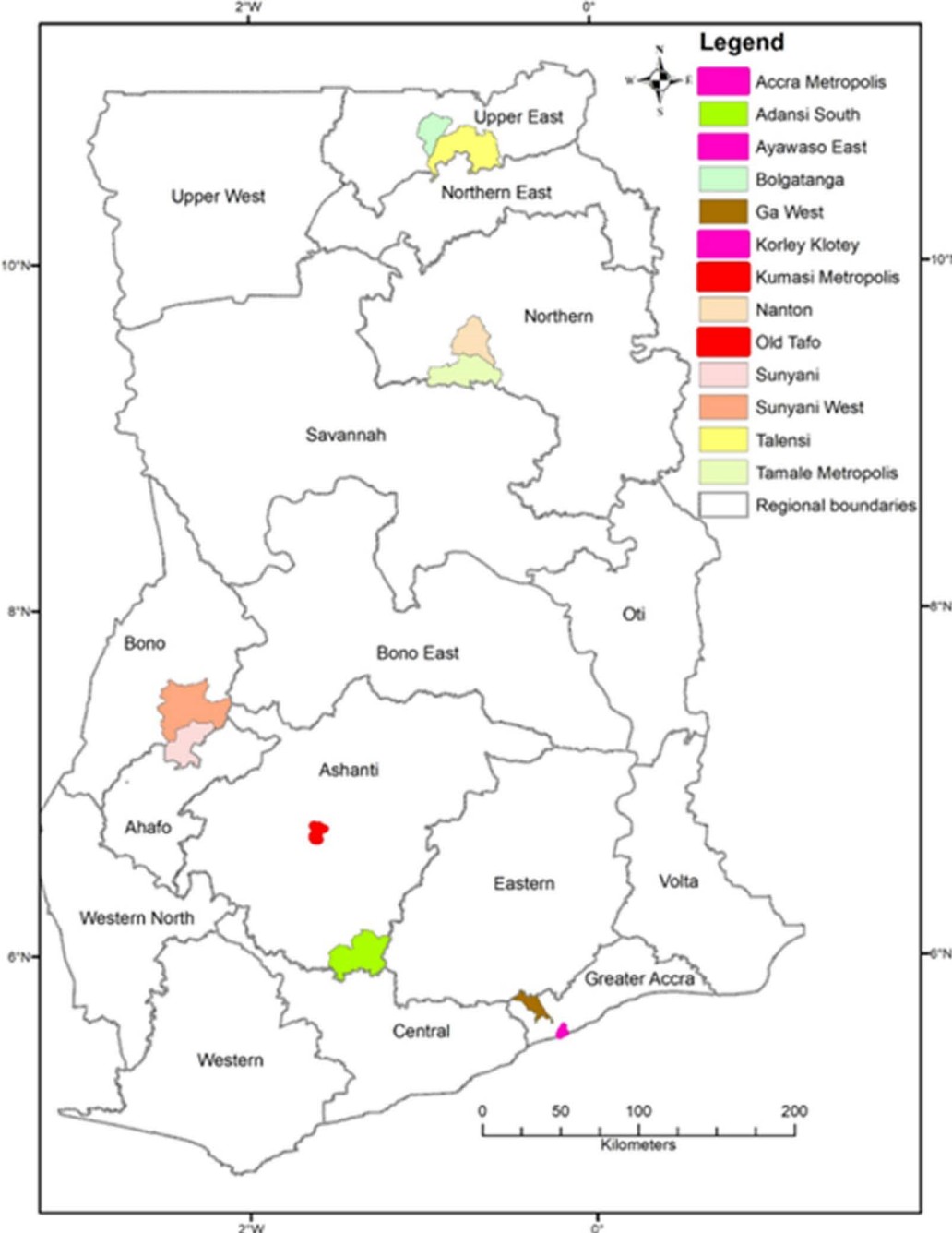

**Fig 1. Map of Ghana showing the study locations.**

parameters considered were (i) the size of the population, (ii) prevalence level, (iii) confidence level of 95% and (iv) 5% accepted deviation from expected prevalence and (v) a non-respondent's rate of 5%. The combined population of the target districts was 956,597 [65]. Being a cross-sectional descriptive study, the sample of this study did not have to satisfy a statistical power.

For the qualitative arm of this study, a purposive sample, ensuring maximum variation, was utilised [66, 67]. This method was appropriate for identifying and selecting research respondents, by virtue of their being especially knowledgeable or expert by lived experience of mental health policy and services, the study subject, as well as reaching the most available stakeholder groups most available and ready to respond [63,66,67].

Maximum variation was ensured to maximise inclusion and to capture unique and diverse experiences and opinions [68]. It also helped to maximise the efficiency and validity of the qualitative data [62,63,66]. In this regard, respondents from each of the three stakeholder categories, ensuring gender disaggregation, were reached and the most available and ready to participate as respondents engaged in the KIIs and FGDs. Each of the three categories of respondents were covered in the KIIs and FGDs.

This made is possible for an adequate representation of a cross-section of respondents, particularly, persons with living with mental health conditions who were stabilised, coherent and articulate and out of freewill, to communicate their experiences, perspectives, and opinions in an articulate, expressive, and reflective manner [69].

The selected individuals were based on initial desk review of available policies and laws on mental health in Ghana and literature in other countries. A list of mental healthcare and general healthcare service providers of healthcare facilities, health service managers and health planning and policy officials, as well as NGOs and CSOs, for each study location was developed and a selected number of them chosen and contacted to consider participating in the Key Informant Interviews (KIIs) and Focus Group Discussions (FGDs). In addition to this, a snowballing approach was adopted, where the respondents who agreed to participate in the interviews and discussions were asked to recommend others who could be sources of information relevant to the study. This also served to confirm the selected list of mental healthcare providers, health service managers and health planning and policy officials, as well as NGOs and CSOs that were to be contacted. Most of the potential respondents recommended for the qualitative data collection were in the list developed to be contacted.

For the persons living with mental health conditions and primary caregivers, the Self-help peer support Groups (SHGs) of persons with mental health conditions and primary caregivers located in the target study locations were used for the key informant interviews and focus group discussions.

For the national level respondents, a list of names of officials responsible of mental health and general health policy and services were listed and approached for the administration of the survey questionnaire and the KIIs and FGDs.

At the district and community level, rural and urban dimensions were used to select district hospitals and communities from which individual respondents were selected for the key informant interviews and focus group discussions. Persons with mental health conditions were reached through the Self-help peer support Groups that BasicNeeds-Ghana (and NGO) works with and supports. Contacts were also made with family members with relatives who have persons living with a mental health condition and under treatment, as well as from nurses from the Community Mental Health Units,

Other self-advocates were contacted based on their public profile and known activities and pronouncements on mental health issues in Ghana. A conscious effort was made to gain representativeness of respondents at the community and national level consultations to reach relevant respondents through a snow-balling approach.

The interviews and focus groups discussions were included at least one respondent of each of the targeted categories of respondents, while ensuring a gender disaggregation. The final number was determined by having reached saturation [70,71]. When no additional new information that was or could be attained from additional or new interviews and or discussions.

Data saturation where a clear pattern of information is realised and where no new information is provided by additional key informant interview or focus group discussions guided the number of KIIs and FGDs conducted [72,73].

## Data collection

A questionnaire, developed using Microsoft 365 Office form, was used to collect the quantitative data.

The questions of the questionnaire were informed by the themes of indicators for *monitoring of the building blocks of health systems* of the WHO *Handbook of Indicators and their Measurement Strategies* [74]. They covered salient points of each of the health system building blocks.

The governance and leadership topic(s) that were explored covered the level of implementation of legislations and policies on the development and integration of mental health into PHC, presence of plans, mental health units and focal persons at the PHC, and involvement of other health care staff in mental health services, private sector involvement, and monitoring and evaluation of mental health care services and integration at the PHC. The qualitative data covered the same topics as the questionnaire which allowed for in-depth exploration of the questions.

To ensure its reliability and validity, the survey questionnaire was refined from a pilot field test that was conducted in the Ledzekuku Municipality in the Greater-Accra Region in July 2020.

Interview and discussion guides made up of open-ended questions based on the WHO handbook for monitoring the building blocks of the health system, just as the questionnaire, were used for the qualitative data collection through KIIs and FGDs. They were field tested and refined before being used for the data collection.

A number of research respondents participated in both completing the questionnaire and in the key informant interviews and key informant interviews. This was particularly for the national level government officials and healthcare facility managers, leaders of SHGs, and disability/ human rights advocates.

## Data collection procedure

The questionnaire was self-completed or interviewer assisted by research assistants recruited and trained to support field data collection. Informed consent was obtained for all respondents. The questionnaires were completed in the English Language or translated from the English Language into the local language of the respondent (Gurune, (Frafra), Dagbani, Twi, (Akan), Ga, Hausa). It took between 45–60 minutes to complete an interview session. Administration of the questionnaires was between 18th August 2020–15th December 2020. A total of 1104 respondents were reached with the questionnaire with 830 (469 female) fully completing them.

The Principal Investigator conducted all the KIIs and moderated the FGDs, with the field data collection assistants serving as interpreters, note-takers and also mobilisation of participants. The interviews were held in English Language and the dominant local language of the locality of the respondents. The KIIs took 45–60 minutes, while the FGDs lasted between one-and-a-half and two hours. After each interview and focus group discussion, a summary was provided for respondents to confirm as a form of participant validation [75].Each of the categories of respondents were represented in the KIIs and FGDs ending when saturation was achieved [76,77]. KIIs, (N = 29) of 29 (9 female) respondents, and FGDs, (N = 12) involving 75 (40 female) respondents, were conducted (Table 1).

Data triangulation [70] on the data collected from the different data collection methods was done during analysis to gain richer understanding of complex processes. This was done by aligning the data to various categories and patterns in other to synthesise the information from the methods and data collection instruments utilised. Data triangulation helped to increase the validity and reliability of the data reducing or eliminating altogether bias and ensuring data saturation. [46–48] The KIIs and FGDs were conducted between 17th August 2020 and 28th February 2021. As part of ensuring reflexivity, trustworthiness and rigour of the study, the investigators adhered to scientific steps and processes to generating empirical knowledge, immersing himself with only the data collected from the field, rather than emotive first-hand knowledge of the existing situation.

## Analysis

StataSE 16 (64-bit) was used to analyse the quantitative data. Descriptive statistical methods were employed to summarise the data into frequencies and percentage across various leadership and governance domains. Thematic analysis was used for the qualitative data analysis with the assistance of NVivo 12 [78–80]. The KIIs and FGDs were recorded on digital devices (voice recorders and phones), transcribed verbatim and coded. The recordings, transcripts and the fieldnotes altogether

PLOS Global Public Health

**Table 1. Respondents of qualitative data collection[1].**

| Region | Method | Stakeholder respondent(s) | | | | | | | | |
|---|---|---|---|---|---|---|---|---|---|---|
| | | Mental Health User | Carer | Health Policy Official/ Health Manager | [Mental] Health Clinician/ Service Provider | Educa-tors | Social Workers/ community volunteers | Mental Health Advocate/ Human Rights workers | Donor/ Dev't partner | Total |
| Upper East Region | KII | 2 (1F) | 2(1F) | 1 | 1 | 0 | 0 | 0 | | 6(2F) |
| | FGD | 2 (1F) | 2(1F) | 0 | 0 | 0 | 0 | 0 | | 4(2F) |
| Northern Region | KII | | | 1 | | | | 1 | | 2 |
| | FGD | | | | | | | | | |
| Bono Region | KII | 2 (1F) | 2 (1F) | 2 | 3(1F) | 0 | 0 | 1 | | 12(3F) |
| | FGD | 2 (1F) | 2 (1F) | 2 | 1F | | | | | 7(3F) |
| Ashanti Region | KII | 2(1F) | 1F | | 1 | | | | | 4(2F) |
| | FGD | | | | | | | | | |
| Greater Accra Region | KII | 1 | | 2 (1) | | | | | 2(1) | 5(2F) |
| | FGD | 1F | | | | | | | | 1F |

[1]'F' is Female. FGDs and KIIs were carried out across the categories of respondents for the study. Stakeholders not included in the districts are those at the national level and not available in the districts.

enabled triangulation of the data obtained. A codebook was initially developed using the conceptual dimensions of the interview guides. After the data collection, samples of the interviews were reviewed and new codes added to the codebook. Double coding was done after which it was compared. The inter-coder reliability index was computed as 0.87, similar to earlier studies [81]. The themes used during the analysis were: (i) the Mental Health Law and level of awareness at the community level; (ii) mental health policy and level of awareness at the community level; (iii) presence of a mental health plan at the district/ sub-district/ CHPS level and level of implementation; (iv) a mental health unit located at sub-district/ CHPS level; (v) coordinator or focal person(s) for mental health; (vi) involvement of healthcare staff in mental health service development; (vii) private sector participation in mental health service delivery; and (viii) monitoring and evaluation of mental health services. The level of integration was based on self-reported perceptions of the three levels of integration provided – (i) fully integrated, (ii) partially integrated, and (iii) not integrated of each the components of each of the health system building blocks, in this case the leadership and governance building block, with an overall impression score.

## Results

A total of 830 study participants completed the survey from a convenience sample of 1010 (response rate of 82%), a majority of them female (56.6%).

Tables 2–6, below provide survey respondents responses to the various themes examined with a final/ overall assessment (Table 6) of participants perspectives of the extent to which the prevailing leadership and governance arrangements fosters integration of mental healthcare at the PHC level. The key areas covered to give indication to the level of stewardship, oversight and coordination were on mental health legislation, policy and plans, focal persons for mental health and involvement of other health personnel, private sector involvement, and monitoring and evaluation of mental healthcare at the PHC.

Generally, mental health leadership and governance at the PHC level was found as low by 48.9% (409/830) of survey respondents, with more than half of the respondents holding such view being health policy officials, health policy implementers and healthcare service providers, 53.1% (137/258), followed by mental health/ human rights advocates, development partners and donors, 48.7% (54/111), and 46.6% (215/461). By geographical location, leadership and governance was low, by 50.5% (419/830) of respondents. Mid Ghana had a high percentage of respondents, 73.6% (201/273)

**Global Public Health**
PLOS

Table 2. Study participants' views on mental health leadership and governance at the PHC level and perception of the level of integration in Ghana by respondent category.

| Health System Block | Leadership & Governance | | | | | | | | | | | | | | | |
|---|---|---|---|---|---|---|---|---|---|---|---|---|---|---|---|---|
| Indicator | Health Policy Officials & Implementers/ Healthcare Service providers | | | | Mental Health Service Users/ Caregivers/ families | | | | Other (Mental Health/ Human Rights Advocates, Health Development Partners/ Donors) | | | | Overall | | | |
| | Integration – (No. (%))[a] | | | | Integration – (No. (%)) | | | | Integration – (No. (%)) | | | | Integration – (No. (%)) | | | |
| | Yes/ Full/ High | Par-tial/ Aver-age | No/ Low/ None Exists | Total | Yes/ Full/ High | Par-tial/ Aver-age | No/ Low/ None Exists | Total | Yes/ Full/ High | Par-tial/ Aver-age | No/ Low/ None Exists | Total | Yes/ Full/ High | Par-tial/ Aver-age | No/ Low/ None Exists | Total |
| Mental health Law & level of awareness at the PHC level | 107 (41.5) | 55 (21.3) | 96 (37.2) | **258 (100)** | 40 (8.7) | 270 (58.6) | 151 (32.8) | **461 (100)** | 54 (48.6) | 30 (3.6) | 27 (24.3) | **111 (100)** | 201 (24.2) | 355 (42.8) | 274 (33.0) | **830 (100)** |
| Level of implementation of the mental health law | 10 (3.9) | 100 (38.8) | 148 (57.4) | **258 (100)** | 103 (22.3) | 141 (30.6) | 217 (47.1) | **461 (100)** | 6 (5.4) | 68 (61.3) | 37 (33.3) | **111 (100)** | 119 (10.5) | 309 (43.5) | 402 (45.9) | **830 (100)** |
| Mental health policy & level of awareness at the PHC level | 20 (7.8) | 150 (58.1) | 88 (34.1) | **258 (100)** | 98 (21.3) | 108 (23.4) | 255 (55.3) | **461 (100)** | 10 (9.0) | 83 (74.8) | 18 (16.2) | **111 (100)** | 128 (12.7) | 341 (52.1) | 361 (35.2) | **830 (100)** |
| Mental health plan at the PHC level | 125 (48.4) | 0 (0.0) | 133 (51.6) | **258 (100)** | 163 (35.4) | 0 (0.0) | 298 (64.6) | **461 (100)** | 56 (50.5) | 0 (0.0) | 55 (49.5) | **111 (100)** | 344 (41.5) | N/A | 486 (58.5) | **830 (100)** |
| Mental health plan in place being implemented at PHC level | 125 (48.4) | 0 (0.0) | 133 (51.6) | **258 (100)** | 163 (35.4) | 0 (0.0) | 298 (64.6) | **461 (100)** | 56 (50.5) | 0 (0.0) | 55 (49.5) | **111 (100)** | 344 (41.5) | N/A | 486 (58.5) | **830 (100)** |
| Mental health unit at the PHC level | 240 (93.0) | 0 (0.0) | 18 (7.0) | **258 (100)** | 393 (85.2) | NA (0.0) | 68 (14.8) | **461 (100)** | 54 (48.6) | 0 (0.0) | 57 (51.4) | **111 (100)** | 687 (82.8) | N/A | 143 (17.2) | **830 (100)** |

[a]number and proportion for each indicator; [c] scoring ranges for levels of integration.

indicating that integration of mental health leadership and governance at the PHC level was low, followed by northern Ghana, 44.3% (109/246). In the contrary, in southern Ghana, it was gauged as highly integrated by 58.5% (182/311).

By the categories of the survey respondents (Tables 2 and 3), majority of health policy officials, healthcare managers and healthcare service providers, 41.5% (107), as well as mental health/ human rights advocates, development partners and donors, by 48.6% (54), had high level of awareness of mental health legislation.

### Level of awareness and implementation of mental health legislation, policy and plan

Awareness of mental health legislation among mental health service users and caregivers was partial with more than half of them, (58.6% [270]), indicating so. Overall, survey respondents' level of awareness of the mental health legislation was partial with 42.8% (355/830) of them indicating so. By geographical location, as presented in Table 4 and 5, general awareness of Ghana's Mental Health Law at the PHC level was high among survey respondents in Northern Ghana, by 48.0% (118) and Mid-Ghana, 44.3% (121), but partial among respondents located in Southern Ghana, 48.2% (150).

Less than a quarter of survey respondents (24.2% [201/830]) were of the view that there were high levels of awareness of the mental health law while 33.0% (274/830) indicated there was a low level of awareness .

KIIs and FGDs with study participants indicated a partial level of awareness of the mental health legislation at the PHC level, particularly among the mainstream mental healthcare workers and health policy officials and advocates and

**Table 3. Study participants' views on mental health leadership and governance at the PHC level and perception of the level of integration in Ghana by respondent category.**

| Health System Block | Leadership & Governance | | | | | | | | | | | | | | | |
|---|---|---|---|---|---|---|---|---|---|---|---|---|---|---|---|
| | Health Policy Officials & Implementers/ Healthcare Service providers | | | | Mental Health Service Users/ Caregivers/ families | | | | Other (Mental Health/ Human Rights Advocates, Health Development Partners/ Donors) | | | | Overall | | | |
| | Integration – (No. (%)) | | | | Integration – (No. (%)) | | | | Integration – (No. (%)) | | | | Integration – (No. (%))) | | | |
| Indicator | Yes/ Full/ High | Partial/ Average | No/ Low/ None Exists | Total | Yes/ Full/ High | Partial/ Average | No/ Low/ None Exists | Total | Yes/ Full/ High | Partial/ Average | No/ Low/ None Exists | Total | Yes/ Full/ High | Partial/ Average | No/ Low/ None Exists | Total |
| Coordinator/ Focal Person at the PHC level | 230 (89.1) | 0 (0.0) | 28 (10.9) | **258 (100)** | 313 (67.9) | NA (0.0) | 148 (32.1) | **461 (100)** | 40 (36.0) | 0 (0.0) | 71 (64.0) | **111 (100)** | 583 (70.2) | N/A | 247 (29.8) | **830 (100)** |
| Involvement of health staff in mental health service development at the PHC level | 20 (7.8) | 115 (44.6) | 123 (47.7) | **258 (100)** | 9 (2.0) | 234 (50.8) | 218 (47.3) | **461 (100)** | 7 (6.3) | 40 (4.8) | 64 (57.7) | **111 (100)** | 36 (4.3) | 389 (46.9) | 405 (48.8) | **830 (100)** |
| Private sector participation at the PHC | 120 (46.5) | 0 (0.0) | 138 (53.5) | **258 (100)** | 106 (23.0) | 0 (0.0) | 355 (77.0) | **461 (100)** | 35 (31.5) | 0 (0.0) | 76 (68.5) | **111 (100)** | 261 (31.4) | N/A | 569 (68.6) | **830 (100)** |
| Monitoring & Evaluation of mental healthcare services at the PHC level | 135 (52.3) | 0 (0.0) | 123 (47.7) | **258 (100)** | 95 (20.0) | 0 (0.0) | 366 (79.4) | **461 (100)** | 50 (45.0) | 0 (0.0) | 61 (55.0) | **111 (100)** | 280 (33.7) | N/A | 550 (66.3) | **830 (100)** |
| Perception of extent of integration of mental health services at the at the PHC level (overall) | 87 (33.7) | 34 (13.2) | 137 (53.1) | **258 (100)** | 188 (40.8) | 58 (12.6) | 215 (46.6) | **461 (100)** | 45 (40.5) | 12 (10.8) | 54 (48.7) | **111 (100)** | 320 (38.6) | 137 (16.5) | 406 (48.9) | **830 (100)** |

members. This was so because there had been mental health law dissemination and sensitisation workshops that they had participated in.

*"I was part of workshop where they told us about a new mental health law, but I really do not know the details of it or how it can help us."* **KII, mental health service user**

*"We were informed about the mental health law in a meeting that we were called to attend at Dapore-Tindongo [a section of Bolgatanga]. At that meeting we were made to understand that government has a responsibility to look after groups like ours in Bolga[tanga] here. That every person living with mental illness wherever located should be assisted by the government. The law requires of government as such. That is how the shared the information with us at the meeting that we attended."* **FGD of mixed group of mental health service users and primary caregivers.**

The qualitative data on awareness of Ghana's mental health legislation at the PHC level corroborates with the results of the field survey, indicating some level of awareness exists, particularly high among health policy officials and health service implementers and among survey respondents located in Southern Ghana, but less so among mental health mental health service users and caregivers, as well as in the northern part of the country.

The extent of implementation of the mental health law was gauged to be low among health policy officials and health service implementers, 57.4% (148/258), and mental health service users and caregivers 47.1 (217/461), and as partial among mental health/human rights advocates, 33.3% (37/111). Overall, the survey participants, by 45.9% (402/830) were of the view that the level of implementation of Ghana's mental health law at the PHC level was low with 43.5% (309/830).

**Table 4. Study participants' views on mental health leadership and governance at the PHC level and perception of the level of integration by geographical zone of the respondents.**

| Health System Block | Leadership & Governance | | | | | | | | | | | | | | | |
|---|---|---|---|---|---|---|---|---|---|---|---|---|---|---|---|---|
| Indicator | Northern Ghana | | | | Mid-Ghana | | | | Southern Coastal | | | | Overall | | | |
| | Integration | | | | Integration | | | | Integration | | | | Integration | | | |
| | Yes/ Full/ High | Par- tial/ Aver- age | No/ Low/ None Exists | Total | Yes/ Full/ High | Par- tial/ Aver- age | No/ Low/ None Exists | Total | Yes/ Full/ High | Par- tial/ Aver- age | No/ Low/ None Exists | Total | Yes/ Full/ High | Par- tial/ Aver- age | No/ Low/ None Exists | Total |
| Level of implementation of the mental health law | 26 (10.6) | 102 (41.5) | 118 (48.0) | **246 (100)** | 50 (18.3) | 118 (43.2) | 105 (38.5) | **273 (100)** | 90 (28.9) | 160 (51.4) | 61 (19.6) | **311 (100)** | 166 (20.0) | 380 (45.8) | 284 (34.2) | **830 (100)** |
| Mental health policy & level of awareness at the PHC level | 20 (8.1) | 100 (40.7) | 126 (51.2) | **246 (100)** | 48 (17.6) | 122 (44.7) | 103 (37.7) | **273 (100)** | 118 (37.9) | 126 (40.5) | 67 (21.5) | **311 (100)** | 186 (22.4) | 348 (41.9) | 296 (35.7) | **830 (100)** |
| Mental health plan at the PHC level | 40 (16.3) | NA (0.0) | 206 (83.7) | **246 (100)** | 101 (37.0) | NA (0.0) | 172 (63.0) | **273 (100)** | 119 (38.3) | NA (0.0) | 192 (61.7) | **311 (100)** | 260 (31.3) | 348 (41.9) | 296 (35.7) | **830 (100)** |
| Mental health plan in place being implemented at PHC level | 40 (16.3) | NA (0.0) | 206 (83.7) | **246 (100)** | 98 (35.9) | NA (0.0) | 172 (63.0) | **273 (100)** | 107 (34.4) | NA (0.0) | 204 (65.6) | **311 (100)** | 245 (29.5) | NA (0.0) | 585 (70.5) | **830 (100)** |
| Mental health unit at the PHC level | 148 (60.2) | NA (0.0) | 98 (39.8) | **246 (100)** | 177 (64.8) | NA (0.0) | 96 (35.2) | **273 (100)** | 285 (91.6) | NA (0.0) | 26 (8.4) | **311 (100)** | 610 (73.9) | NA (0.0) | 220 (26.5) | **830 (100)** |

**Table 5. Study participants' views on mental health leadership and governance at the PHC level and perception of the level of integration by location of the respondent.**

| Health System Block | Leadership& Governance | | | | | | | | | | | | | | | |
|---|---|---|---|---|---|---|---|---|---|---|---|---|---|---|---|---|
| Indicator | Northern Ghana | | | | Mid Ghana | | | | Southern Ghana | | | | Overall | | | |
| | Integration | | | | Integration | | | | Integration | | | | | | | |
| | Yes/ Full/ High | Par- tial/ Aver- age | No/ Low/ None Exists | Total | Yes/ Full/ High | Par- tial/ Aver- age | No/ Low/ None Exists | Total | Yes/ Full/ High | Par- tial/ Aver- age | No/ Low/ None Exists | Total | Yes/ Full/ High | Par- tial/ Aver- age | No/ Low/ None Exists | Total |
| Coordinator/ Focal Person at the PHC level | 148 (60.2) | NA (0.0) | 98 (39.8) | **246 (100)** | 179 (65.6) | NA (0.0) | 94 (34.4) | **273 (100)** | 287 (92.3) | NA (0.0) | 24 (7.7) | **311 (100)** | 614 (74.0) | NA (0.0) | 216 (26.0) | **830 (100)** |
| Involvement of health staff in mental health service development at the PHC level | 34 (13.8) | 101 (41.1) | 111 (45.1) | **246 (100)** | 67 (24.5) | 99 (36.3) | 107 (39.2) | **273 (100)** | 97 (31.2) | 100 (32.2) | 114 (36.7) | **311 (100)** | 198 (23.9) | 300 (36.1) | 332 (40.0) | **830 (100)** |
| Private sector participation at the PHC | 48 (19.5) | 92 (37.4) | 106 (43.1) | **246 (100)** | 71 (26.4) | 40 (14.7) | 161 (59.0) | **273 (100)** | 109 (35.0) | 112 (36.0) | 90 (28.9) | **311 (100)** | 229 (27.6) | 244 (29.4) | 357 (43.0) | **830 (100)** |
| Monitoring & Evaluation of mental healthcare services at the PHC level | 31 (12.6) | 92 (37.4) | 106 (43.1) | **246 (100)** | 64 (23.4) | 69 (25.3) | 140 (51.3) | **273 (100)** | 91 (29.3) | 112 (36.0) | 90 (28.9) | **311 (100)** | 186 (22.4) | 228 (27.5) | 416 (50.1) | **830 (100)** |

**Table 6. Mental health services at the PHC and their level of integration by Respondent category and Geographical Zone (N = 830).**

| Indicator | Respondent Category | | | | | Respondent Geographical Zone | | | | |
|---|---|---|---|---|---|---|---|---|---|---|
| | Health Policy officials/ Health Managers/ Providers | Persons with MH Conditions/ Caregivers | 'Others' | 95% CI | P-Value | Northern Ghana | Mid-Ghana | Southern Ghana | 95% CI | P-Value |
| Coordinator/ Focal Person at the PHC level | 2.40 | 2.80 | 2.60 | 2.30 - 3.00 | 0.032 | 2.30 | 2.60 | 3.10 | 2.10 - 3.20 | 0.045 |
| Involvement of health staff in mental health service development at the PHC level | 2.60 | 2.40 | 2.70 | 2.20 - 2.90 | 0.041 | 2.50 | 2.20 | 2.90 | 2.00 - 3.10 | 0.037 |
| Private sector participation at the PHC | 2.20 | 2.70 | 2.40 | 2.00 - 2.90 | 0.038 | 2.10 | 2.50 | 2.80 | 1.90 - 3.00 | 0.048 |
| Monitoring & Evaluation of mental healthcare services at the PHC level | 2.90 | 2.50 | 2.70 | 2.30 - 3.00 | 0.035 | 2.80 | 2.40 | 3.00 | 2.20 - 3.30 | 0.042 |
| Overall mean Integration | 2.53 | 2.60 | 2.60 | 2.30 - 3.20 | 0.033 | 2.43 | 2.43 | 2.95 | 2.20 - 3.30 | 0.038 |

NB: Mean value >3.0 = Full integration; Mean value of 2.0-3.0 = Partial integration; Mean value <2.0 signifies No integration.

The p-values in the tables indicate the statistical significance of the observed differences across the groups. Lower p-values (<0.05) suggest stronger evidence that the observed differences in mean values are less likely due to random chance.

With regards to the geographical location of the study respondents, the level of implementation of the mental health law was low among respondents of northern Ghana, by 48.0% (118/246), and average for respondents in mid Ghana, 43.2% (118/273), and southern Ghana, 51.4% (160/311). Overall, the level of implementation of the mental health law was partial, 45.8% (380/830), followed by 34.2% (284/830) of survey respondents whose views were that it was low. Results of qualitative data indicate the level of implementation between partial and low as the quotes below conveys.

*"Mental Health Act promoting community-based mental health services - It does promote. Very good. On paper it does promote. In practice, we need to get the structures called for by the Law and that is where I am saying that we need to get the district coordinators in place, which we haven't got in place yet. So, it leads to a certain gap. We have the regional coordinators they are there, and they are functioning very well, but the district coordinators need to be put in place."* **KII, Health Policy Official**

*"There is a mental health law and the NGOs working in mental health and the MHA have done a lot to publicise it, but the law is limited in its compliance with the UNCRPD provisions, particularly the aspect of involuntary treatment. It is at variance with the [UN] CRPD and not upholding human rights of people with mental health and psychosocial disabilities."* **KII, mental health advocate.**

Mental health policy and level of awareness at the PHC level was gauged to be partial among mental health/ human rights advocates, (74.8% [83/111]), and health policy officials and implementers, (58.1% [150/258]), and low among mental health service users and care givers by 55.3% (255/461). Less than ten percent of health policy official and health service implementers (7.8% 20/258]), and mental health/ human rights advocates, (9.0% [10/111]) indicated awareness of Ghana's mental health policy and its level of implementation, but almost a quarter (21.3% [98/461]), of mental health service users and caregivers of the view that there was high awareness of the policy and its implementation. Generally, among the three major categories of survey respondents, the majority of them were of the view that mental health policy and implementation at the PHCe level was partial, 52.1% (341/830), with 35.2% (361/80) indicating a low level of awareness and implementation.

By the geographical location of the respondents, the awareness of mental health policy and its level of implementation at the PHC was partial among respondents of all the three locations, 44.7% (122/273) mid Ghana, 40.7% (100/246) from

northern Ghana and 40.5% (126/311) in southern Ghana. Overall, awareness of Ghana's mental health policy and its implementation at the PHC level was partial, (41.9% (348/830)) across the three geographical zones, followed by 35.7% (296/830) of respondents indicating low or non-implementation, with less than a quarter, (22.4% (186/830)) of the view that there was a high level of awareness and implementation of the mental health policy at the PHC level.

Survey respondents' views about the level of implementation of the mental health policy and law are about similar at between 41.9% (348/830) for mental health policy and 45.8% (380) of survey respondents by geographical location. It is wider among the three broad categories of respondents, being between 43.5% (309/830) of the respondents who assessed it as partial with regards to awareness and implementation of mental health policy at the PHC level, and 52.1% (341/830) for mental health law implementation within the geographical locations that the survey respondents are located in. Qualitative data mirror responses from the qualitative data collected.

*"Well, I think it* [*the Mental Health Law*] *actually touches the points that are needed to be touched on, but as usual, Ghana, we come out with fine laws, fine policies but the implementation on the ground leaves much to be desired. For often for all you know, the law promotes mental health activities and mental health service at the community level."* **KII, District Director of Health Services**

The presence of a mental health policy and legislation emphasising mental healthcare to be community-based integrated into general care at the PHC level along *clinical integration* (coordination of person-focused care in a single process across time, space and discipline,

### Presence of focal persons for mental health at the PHC and level of involvement of health workers in mental health at the community level

There was also interest in knowing if there was a mental health plan in place at the PHC level and if it was implemented at that level. Similar responses of awareness and implementation were provided. More than half (51.6% [133/258]) of health policy officials and health service implementers category indicated that no mental health plan at the PHC level. Among the category of mental health service users and caregivers, 64.6% (298/461) of them indicated there was no mental health plan in place and or being implemented at the PHC level. For mental health/ human rights advocates however, half of them (50.5% [56/111]) answered in the affirmative that a mental health plan was in place at the PHC level and being implemented. With regards to geographical location of the respondents, 83.7% (206/246) of survey respondents located in northern Ghana, 63.0% (172.273) in mid Ghana and 61.7% (192/311) from the southern Ghana answered that there was no mental health plan at the PHC level or implemented. Qualitative data collected, however, conveyed that beyond the weekly and monthly plan of activities that some of the mental health workers draw to work with there is no annual or strategic plan in place to guide the delivery of mental healthcare services at the district and lower levels of the healthcare system of Ghana.

*"We don't have plans. Plan of activities depends on finances and since we do not have, we just do the bit we can – some consultations in the hospital for those who come and occasional home visits. We only developed plans when NGOs like BasicNeeds* [-Ghana] *come to help us or the Mental Health Authority, if they get funding from DFID and channel some to us to work with."* **FGD, CPNs/CMHOs,**

The presence of a mental health unit at the PHC is an important indication of a sense of structure and leadership of mental healthcare at that level. Along with interest in knowing if mental health units were present at the PHC level was to know whether there were designated coordinators for mental healthcare at that level. Majority of the respondents answered in the affirmative by 82.8% (687/830) for presence of a mental health unit and 70.3% (583/830) for presence of a coordinator or focal person for mental health at the PHC level. Of these, the majority of respondents were health policy officials and

health service implementers, 93.0% (240/458) for presence of a mental health unit and 89.1% (230/258) for presence of a mental health coordinator or focal person. Similar responses were provided by mental health service users and caregivers, by 85.2% (393/461) for presence of a mental health unit and 67.9% (313/461) for presence of a focal person. However, the difference in percentage of responses indicating the presence of a mental health unit with that for presence of a mental health coordinator/ focal persons could be indication of units that have no personnel manning them. Geographically, affirmative responses to the presence of mental health units and mental health coordinators at the PHC level were lower for respondents in northern Ghana, 60.2% (148/246), higher in mid Ghana, 64.8% (177/273), and highest in, 91.6% (285/311), for respondents located in southern Ghana.

Responses of the qualitative data collected conveyed mixed views as below.

*"At least there is a CMHO in the majority of districts of Ghana but not below. The challenge is the tools and resources for them to work with. I mean funds, logistics like motorbikes, proper offices, and furniture. If these are provided, we can go to the sub-districts and into the hinterlands to provide services."* **KII, regional mental health coordinator.**

*"We do not have a unit in our clinic, we have to go to the mental health unit at the regional hospital."* **FGD, Service users and caregivers.**

With regards level of involvement of health staff at the PHC level of the healthcare system in mental health service development, a majority of the three categories of survey respondents, 48.8% (405/830) were of the view that involvement of healthcare staff was low followed by 46.9% (389/830) who answered that there was a partial level of involvement of health staff in mental health service development and delivery. Just 4.3% (36/830) answered that there was a high level of involvement of health staff in mental healthcare development. Of categories of respondents, mental health and human rights advocates, development partners and donors were the majority, 57.7% (64/111) who indicated that there was a low level of involvement of health staff in mental healthcare service development at the PHC level. Health policy officials and health service implementers, on one hand and mental health service users and caregivers, on the other, respectively, by 47.7% (123/258) and 47.3% (218/461) expressed that there was low involvement of other health staff in the development of mental health service at the PHC level in Ghana. Just 4.3% (36/830) indicated that healthcare staff are fully involved in mental healthcare service development at the community level.

By geographical location, respondents located in northern Ghana were the majority, 45.1% (111/246), who indicated that there was low level of involvement of health personnel at the PHC level. They were followed by respondents from mid Ghana, 39.2% (107/273) and southern Ghana, 36.7% (114/311). Overall, by geographical location, survey respondents viewed the level of involvement of health staff in mental health service development at the PHC as low, 40.0% (332/830) or partial, 36.1% (300/830).

Qualitative data corroborate the survey results as the quote below convey:

*"Unless you are working in mental health, healthcare workers are not interested in mental health matters. There is fear and social stigma. It is only the CPNs [Community Psychiatric Nurses] and the CMHOs [Community Mental Health Officers] who do everything. A negligible number are really interested and involved."* **KII, District Director of Health Services**

## Private sector involvement in mental healthcare at the PHC level

There was also interest in understanding of private sector participation in the development and delivery of mental healthcare services at the PHC level. By the three categories of survey respondents, 77.0% (355/461) of mental health service users and caregivers indicated that was not in place. The category of mental health and human right

advocates, development partners and donors, by 68.5% (76/111) similarly indicated that private sector participation in mental health care delivery at the PHC level was not in place, just was the case with 53.5% (138/258) health policy officials and health service implementers. For respondent who answered in the affirmative, their reference was the presence of traditional and faith-based healers who are highly patronised by people with mental healthcare needs. The results of the data analysed by geographical location of the respondents, an overall of 43.0% (357/830) of respondents were of the view that private sector participation in mental healthcare services was low, with 59.0% (161/273) being those located in mid Ghana, followed by respondents from northern Ghana, 43.1% (106/246), and lastly by those located in the southern parts of Ghana, 28.9% (90/311). This could be attributed to the presence of private healthcare facilities in the urban areas. In the case of respondents who were of the view that there was no private sector involvement in the delivery of mental health services at the PHC level, there was absence of private clinics and hospitals providing mental healthcare services. Qualitative data analysed corroborate results of the survey, conveyed in the quotes below:

> *"The services provided by the traditional healers makes me say that there is private sector participation in mental healthcare service delivery in Ghana. There are also a number of the psychiatrists and churches with rehabilitation centres that I have come to be aware of."* **KII, Mental health policy official**

> *"For us here, it is only the psychiatry unit that is at the regional hospital. We have many private hospitals and maternity homes, but none offer mental healthcare services. It is only at the government hospitals that mental healthcare services are provided."* **FGD, Mental health service users**

### Monitoring and Evaluation of mental healthcare at the PHC

Monitoring and evaluation form an important aspect of healthcare stewardship. Monitoring and Evaluation of mental healthcare services at the PHC level was also gauged to understand the level of leadership and governance of mental healthcare at the community level. A majority, 66.3% (550/830), of all the three categories of survey respondents combined were of the view that monitoring and evaluation of mental health care at the PHC level was low. The highest percentage of survey respondents who viewed mental healthcare monitoring and evaluation to be low were the category of mental health service users and caregivers, 79.4% (366/461), followed by mental health/ human rights advocates, development partners and donors, 55.0% (61/111) and finally, health policy officials and health care service providers/ implementers, 47.7% (123/258). Results of analyses of the survey data by the geographical location of the respondents had an overall of 50.1% (416/830) of the view that monitoring and evaluation at the PHC level was not in place. Respondents of mid-Ghana constituted the highest percentage of respondents, 51.3% (140/273), who gauged monitoring and evaluation of mental healthcare at the PHC to be low, followed by respondents located in northern Ghana, 43.1% (106/246). However, respondents of southern Ghana viewed monitoring and evaluation at the PHC level to be partial. Qualitative data, as per the quote below lends credence to the results of the survey contained in the quote below:

> *"It was once that staff of the Mental Health Authority at the headquarters and the directors of the Institutional Care Division (ICD) of the Ghana Health Service undertook supportive supervision but that was just to the regional hospital. Also, once a while the Regional Mental Health Coordinator comes around but that is far and in-between."* **FGD, CPNs/ CHMOs**

> *"Mental health is not considered in matters at this level. supportive supervisions take place, but mental health is not included. We are just sitting and watching. They may include you, but it is not because of mental health but that you an available staff."* **KII, CPN**

Respondents self-reported perception of leadership and governance based on the selected themes were gauged as low, by 48.9% (406/830),A significant percentage also gauged it as high (38.6%). Close percentages of mental health service users and caregivers and mental health and human rights advocates, development partners and donors were divided in the views with 40% and more of the view that leadership and governance was in place at the PHC and litte more than 40% guaging leadership and governance not in place

*"Leadership is evident when the district director is interested in what is happening with the mental health aspects of his district. Without this level of involvement, we should forget discussing leadership and governance at the district and lower levels."* **KII, Mental Health Advocate**

*"Mental health remains an afterthought and is excluded so leadership and governance is remote, not there. It needs to be consciously crafted and health directors accountable."* **KII, National Health Policy**

*"It is sometimes confusing. You do not know where you belong, who you should be reporting to. So, to be save you report to the* [District] *Director and then copy the regional mental health coordinator. There hardly any coordination, monitoring or supervision."* **FGD, CPNs and CMHOs**

*"We cannot say there is no leadership and governance of mental health in place at the Primary Care level. Increasingly, directors are taking an interest and through that the situation will improve."* **KII, Regional Director of Health Services**

### Overall perception of the level of leadership and governance at the PHC

Composite mean scores on leadership and governance for the indicators used to assess the level of integration further elucidate the study's results of the situation. Overall, level of integration of leadership and governance for mental health-care service integration at PHC level was assessed to be partial with the lowest means scores being among the category of health policy providers, healthcare facility managers and healthcare service providers (2.53) and then by 2.60 respectively among the category of mental health care service users, primary caregivers and families, and the category of human rights/ patient rights advocates, development partners and donors. Respondents from Northern Ghana and Mid-Ghana zones had a lower score, (2.43), with Southern Ghana with a higher assessment (2.95) of the partial level of leadership and governance for mental health at the PHC.

## Discussions

Legislations, policies and plans are elements of and can be considered proxies of good governance and leadership structures. Healthcare policies are described as purposeful and deliberate actions whose efforts are made to strengthen health systems and changes within the health system(s) in order to promote population [82–84]. Legislation on the other hand serves as declaration of policy, as a statement of principle or social ideal to create and promote social values. Health policies and legislations therefore play a critical role in health system strengthening and influence societal relations in addressing the social determinants of health and health inequities. Availability of healthcare-related policies and legislations and the implementation constitute a key part of the WHO health systems framework. From this study, there are adequate policies and legislations that support the development of mental healthcare services at the community level. The interest in this study was not about how healthcare policies and legislations are developed but more of how they are prioritised and awareness of them among a cross-section of relevant stakeholder groups of the general population. That gives indication of the level of attention that should paid to the inequalities and inequities existing within mental healthcare service sector in Ghana.

The mental health law and the national mental health policy of Ghana were presented to seek respondents' awareness of them and if they were being implemented as stipulated using a survey instrument and KIIs and FGDs.

Overall, mental health leadership and governance of mental healthcare at the district and lower levels of the healthcare system of Ghana was average. In this study general awareness of the mental health law and policy was high. The level of implementation was similarly average. Mental health units and mental health coordinators or focal persons were present but their level of involvement in healthcare service planning and delivery low making mental health siloed. Private sector participation was low, and when it was identified to exists, it was in reference to the activities of traditional and spiritual healers. This is an indication of mental health gaining a gradual but steady recognition with the need for mental healthcare to be decentralised and integrated into general healthcare at the PHC level as the obvious and most effective option. This generally average level of awareness gives indication that more needs to be done for the provisions of the mental health law and policy to be not just known about but fully implemented. As an organogram, the structure of mental healthcare at the primary care level is reasonably well established except with variable coverage and quality of the service, which is due to inadequacies of trained human resources and logistics and cross-sectoral linkages. It is one thing to know about the mental health law but is more important to know that the law is being implemented. This gives indication that stakeholder involvement in healthcare related policy issues and legislations needs to be stepped up.

Level of awareness of mental health legislation was affected by the broader social determinants of mental health which was demonstrated in the socio-demographic characteristics of the respondents, particularly geographical locations. The study found that there was least awareness of the mental health law and policy among respondents of the northern zone and high among those in the southern-coastal. The mid Ghana zone was next, lower than the southern coastal zone but higher than the northern zone. Challenges with realising the full implementation of the mental health Law (and policy) have been previously highlighted [85,86] as a key leadership and governance militating against the development of mental healthcare services in Ghana. From this study, mental health leadership and governance at the community level in Ghana remains low. Mental health service users and primary caregivers of persons with mental health conditions who were members of Self-Help peer support Groups (SHGs) that NGOs, such as BasicNeeds-Ghana, as well as mental health advocates exhibited better grasp of existing mental-health related policies and legislations than the respondents of similar characteristics who were not. This is a result of the knowledge and skills capacity strengthening activities and empowerment initiatives undertake to enhance participation of mental healthcare service users and their caregivers [87].

This study established that awareness of the mental health law and policy was partial for which more needs to be done to publicise them especially among mental health services users and primary caregiver and general healthcare service workers. Also, the existing institutional arrangements are overlapping, if not confusing, for focal persons, with regards who they are accountable to, and such a situation contribute to inefficiencies as can be observed. Despite units being at the district level health facilities they are siloed, and district focal persons are both reporting to MHA and GHS.

## Generalisability, limitations, de-limitation

The study's findings are generalisable, and they reflected minimum requirements of a scientific investigation. The study employed a mixed method study design to investigate the integration at the PHC level from sites that reflected a national character of the country. Robust scientific enquiry measures were applied and adhered to which made the findings sound and generalisable.

There were limitations too. First, the study was carried out within a milieu with unique contextual issues, including bureaucracies across the country. As a result, not all the relevant respondents who should participate in the research were engaged. The study delved into investigating the subject of interest within a limited coverage which could lead to some important information that could enrich the study to be missed. Purposive sampling has limitations in identifying people with foreknowledge in the subject matter. To address this limitation, steps were taken to select individuals from institutions that have stake in mental health leadership and governance with the use of policy and stakeholder analysis. The use

of data triangulation methods strengthened from both the quantitative and qualitative components of the study. Financial and time constraints did not make it possible for a such large-scale approach to be applied in the study. Most of the respondents, particularly the people with mental health conditions and primary caregivers were members of self-help peer support groups that BasicNeeds-Ghana, an NGO works for which there must have been quite a good amount of exposure and information that they have. this helped in addressing the depth at which they study could have gone. COVID-19 and the restrictions and protocols to contain its spread and negative impact on people and society in general had an effect on community engagements. However, the required preventive measures were adhered to, and this significantly reduced any negative impact of the situation on the study.

The limitations mentioned herein present an opportunity for further exploration of the subject of integration of mental healthcare at the lower levels of healthcare service system as well as studies on health system strengthening. These factors could be used to develop an intervention to improve the system strengthening needs of te mental health system of Ghana and other Low Middle Income Countries.

## Conclusions

The right to health must be met holistically and every effort ought to be marshalled to, at anytime to enable the majority in need of mental health care services to benefit.

The contribution of mental health to the burden of disease, both infectious diseases and non-communicable diseases, requires innovative responses and sustainable investment in the core building blocks of the health system framework and which needs to be recognised and prioritised in Ghana. Ghana has a mental health law that was passed in 2012 [88–90] while a mental health policy was launched in 2021. Appropriate policies and legislations are key requirements of the health system framework. This notwithstanding, it is not enough to have policies and legislations. There needs to be nationally driven mental health programme backed by resource outlays, particularly funding. A national mental health programme needs an appreciation of the social determinants on mental health which address geographical differences and related socio-economic characteristics.

Political commitment to integration of mental healthcare in general healthcare at the PHC level very necessary. The importance of political skills in fostering mental health reform has been well documented and this needs to be emphasised in Ghana. Being isolated, least prioritised, and even antagonised sometimes, nimble pollical manoeuvering are needed to build support and sustain commitment for mental healthcare, especially at the community level. Political commitment to a process, taking advantage of windows of opportunities helps in navigating the intricacies of the mental health sector. There should a greater focus on leadership and governance arrangements that understand the motivations of the range of stakeholders all levels and the situational demands, and using knowledge and understanding in ways that instill confidence in the system and investment in it. This needs to be done with the urgency that can be mustered for which a new/additional breed of leadership innovative and using political sophistications to promote development of community-based mental health. It will be most helpful for a good cadre of young and highly qualified professionals attracted to work in mental health and along with the elderly experienced ones to bring the much-needed change in the sector.

Clarity is need on the agency to lead mental healthcare services at the community level. The responsible agency for developing mental healthcare service provision remains unclear. The boundaries of authority and responsibility for the development and management of mental healthcare services at the lower levels of the healthcare system of Ghana is blurred which is affecting the pace at which mental healthcare at the community level should be developed. This situation can be traced to the current legislations on mental health (Act 846; 2012) and the Ghana Health Service (Act 525; 1996).

## Supporting information

**S1 Text. Study qualitative data.**
(DOCX)

**S1 Data.  Study quantitative data.**
(XLS)

**S1 Checklist.  PRISMA Checklist.**
(DOCX)

## Acknowledgments

The authors wish to acknowledge Prof. Patricia Akweongo, Prof. Moses Aikins, and Dr Leonard Batiema for their support and advice to the lead author (Peter Badimak Yaro) in his doctoral studies.

## Author contributions

**Conceptualization:** Peter Badimak Yaro, Emmanuel Asampong, Paulina Tindana.

**Data curation:** Peter Badimak Yaro, Philip Teg-Nefaah Tabong, Graham Thornicroft, Paulina Tindana.

**Formal analysis:** Peter Badimak Yaro, Emmanuel Asampong, Philip Teg-Nefaah Tabong, Graham Thornicroft, Paulina Tindana.

**Funding acquisition:** Peter Badimak Yaro.

**Investigation:** Peter Badimak Yaro.

**Methodology:** Peter Badimak Yaro, Emmanuel Asampong, Philip Teg-Nefaah Tabong, Graham Thornicroft, Paulina Tindana.

**Project administration:** Peter Badimak Yaro.

**Supervision:** Emmanuel Asampong, Graham Thornicroft, Paulina Tindana.

**Writing – original draft:** Peter Badimak Yaro.

**Writing – review & editing:** Peter Badimak Yaro, Emmanuel Asampong, Philip Teg-Nefaah Tabong, Graham Thornicroft, Paulina Tindana.

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
