## [Decision Letter · Decision Letter 0]

12 Jan 2024

PGPH-D-23-02157

Leadership and governance of mental healthcare and integration at the community level: a mixed methods study in Ghana

Dear Dr. YARO,

Thank you for submitting your manuscript to PLOS Global Public Health. After careful consideration, we feel that it has merit but does not fully meet PLOS Global Public Health’s publication criteria as it currently stands. Therefore, we invite you to submit a revised version of the manuscript that addresses the points raised during the review process.

Specifically, please provide further details around the process of actor mapping, including identification of relevant stakeholders and definition of the selection criteria. Please provide more details regarding the identification of themes within the qualitative data, contextualize the integration of mental health within the broader healthcare kandscape, and discuss the implications of your research findings, including the limitations. 

We look forward to receiving your revised manuscript.

Kind regards,

Jennifer Tucker, PhD

Staff Editor

Journal Requirements:

2. Please describe in your methods section how capacity to provide consent was determined for the participants in this study. Please also state whether your ethics committee or IRB approved this consent procedure. If you did not assess capacity to consent please briefly outline why this was not necessary in this case.

3. Please provide separate figure files in .tif or .eps format only and remove any figures embedded in your manuscript file. Please also ensure all files are under our size limit of 10MB.

4. We have noticed that you have a list of Supporting Information legends in your manuscript. However, there are no corresponding files uploaded to the submission. Please upload them as separate files with the item type 'Supporting Information'. 

5. Please amend your Data Availability Statement and indicate where the data may be found.

6. Some material included in your submission may be copyrighted. According to PLOS’s copyright policy, authors who use figures or other material (e.g., graphics, clipart, maps) from another author or copyright holder must demonstrate or obtain permission to publish this material under the Creative Commons Attribution 4.0 International (CC BY 4.0) License used by PLOS journals. Please closely review the details of PLOS’s copyright requirements here: PLOS Licenses and Copyright. If you need to request permissions from a copyright holder, you may use PLOS's Copyright Content Permission form.

Potential Copyright Issues:

Fig 1: please (a) provide a direct link to the base layer of the map (i.e., the country or region border shape) and ensure this is also included in the figure legend; and (b) provide a link to the terms of use / license information for the base layer image or shapefile. We cannot publish proprietary or copyrighted maps (e.g. Google Maps, Mapquest) and the terms of use for your map base layer must be compatible with our CC-BY 4.0 license. 

"

Additional Editor Comments (if provided):

Reviewers' comments:

Reviewer's Responses to Questions

**Comments to the Author**

1. Does this manuscript meet PLOS Global Public Health’s publication criteria ? Is the manuscript technically sound, and do the data support the conclusions? The manuscript must describe methodologically and ethically rigorous research with conclusions that are appropriately drawn based on the data presented.

Reviewer #1: Partly

2. Has the statistical analysis been performed appropriately and rigorously?

Reviewer #1: Yes

3. Have the authors made all data underlying the findings in their manuscript fully available (please refer to the Data Availability Statement at the start of the manuscript PDF file)?

Reviewer #1: Yes

4. Is the manuscript presented in an intelligible fashion and written in standard English?

Reviewer #1: Yes

5. Review Comments to the Author

Reviewer #1: December 24, 2023.

Dear Editor and Authors,

I reviewed the article “Leadership and governance of mental healthcare and integration at the community level: a mixed methods study in Ghana.” I would like to express my appreciation for your valuable contribution to the governance field.

However, as I reflected on your article, I believe that enhancing the scientific rigor, especially in the qualitative aspect, could further strengthen your findings. I would like to offer some suggestions for consideration:

Even though the article adeptly outlines the research design and methods, it is crucial to consider certain refinements to ensure methodological rigor. Specifically, in the context of assessing leadership and governance in mental healthcare, a more detailed explanation of the mapping of actors would enhance the clarity and robustness of the study.

Further to strengthen the research design, it is recommended to explicitly mention the process of actor mapping, including the identification of relevant stakeholders and the definition of selection criteria. This will provide readers with a clear understanding of how the study identified and engaged with key actors in mental healthcare at the community level.

Authors should provide more information about how themes or patterns were identified in your qualitative data. This could involve a discussion on the coding process, inter-rater reliability, or any measures taken to ensure the trustworthiness of your qualitative findings.

In reviewing the qualitative methodology in the manuscript, it became apparent that there are some technical discrepancies, particularly regarding participant selection. In qualitative research, participant selection is typically guided by specific criteria rather than random sampling.

I recommend explicitly stating the criteria employed in participant selection, such as specific roles, experiences, or characteristics that make individuals eligible for inclusion in the study. This information is crucial for readers to understand the rationale behind participant selection and ensures that the qualitative data collection aligns with the research objectives.

"Theoretical Saturation" is a concept mainly associated with the qualitative research methodology called "Grounded Theory". How did you adapt it to your method used in your analysis?

Other points to strengthen in your manuscript are the following:

To contextualize the integration of mental health within the broader healthcare landscape, it is essential to draw an outline of the structure of health services in Ghana.

When examining mental health resources and services, it is crucial to utilize a standardized metric for comparison. In this context, I recommend employing the number of resources per 100 thousand inhabitants as a key indicator. This metric allows for a more equitable and insightful assessment of mental health support across different regions or populations. (Lines 94-102 in their manuscript).

Moreover, it would be beneficial to include a brief section discussing the implications of the findings and specific recommendations derived from the study. Also include a section on study limitations and future directions for a more comprehensive discussion.

Finally, when reviewing the manuscript, I observed that two different citation styles are employed, leading to inconsistencies in the presentation of references. Maintaining a uniform citation style is crucial for the overall coherence and professionalism of the article.

6. PLOS authors have the option to publish the peer review history of their article (what does this mean? ). If published, this will include your full peer review and any attached files.

**Do you want your identity to be public for this peer review?** For information about this choice, including consent withdrawal, please see our Privacy Policy .

Reviewer #1: **Yes: ** Dr. Lina Diaz-Castro Ph.D., M.D., Psychiatr.

---

## [Decision Letter · Decision Letter 1]

12 Jan 2025

PGPH-D-23-02157R1

Leadership and governance of mental healthcare and integration at the community level: a mixed methods study in Ghana

Dear Dr. Peter Badimak Yaro, 

Thank you for submitting your manuscript to PLOS Global Public Health. After careful consideration, we feel that it has merit but does not fully meet PLOS Global Public Health’s publication criteria as it currently stands. Therefore, we invite you to submit a revised version of the manuscript that addresses the points raised during the review process.

We look forward to receiving your revised manuscript.

Kind regards,

Roopa Shivashankar, MD, MSc

Academic Editor

Journal Requirements:

Additional Editor Comments (if provided):

Reviewers' comments:

Reviewer's Responses to Questions

**Comments to the Author**

1. If the authors have adequately addressed your comments raised in a previous round of review and you feel that this manuscript is now acceptable for publication, you may indicate that here to bypass the “Comments to the Author” section, enter your conflict of interest statement in the “Confidential to Editor” section, and submit your "Accept" recommendation.

Reviewer #2: (No Response)

Reviewer #3: All comments have been addressed

2. Does this manuscript meet PLOS Global Public Health’s publication criteria ? Is the manuscript technically sound, and do the data support the conclusions? The manuscript must describe methodologically and ethically rigorous research with conclusions that are appropriately drawn based on the data presented.

Reviewer #2: Partly

Reviewer #3: Yes

3. Has the statistical analysis been performed appropriately and rigorously?

Reviewer #2: N/A

Reviewer #3: Yes

4. Have the authors made all data underlying the findings in their manuscript fully available (please refer to the Data Availability Statement at the start of the manuscript PDF file)?

Reviewer #2: Yes

Reviewer #3: Yes

5. Is the manuscript presented in an intelligible fashion and written in standard English?

Reviewer #2: Yes

Reviewer #3: Yes

6. Review Comments to the Author

Reviewer #2: Aligned with the WHO Comprehensive Mental Health Action Plan 2013–2030 and the Sustainable Development Goals, transforming attitudes, approaches, and actions to promote and protect mental health while providing care for those in need is imperative. Establishing community-based mental health services and strengthening collaborative efforts to integrate mental health into primary healthcare systems through strategic leadership and governance are essential components of this transformation. The Mental Health Atlas 2020 highlights significant gaps in leadership and governance—such as inadequate policies, plans, and laws, as well as misplaced priorities—that can critically undermine a country’s mental health response. In this context, the current manuscript, which explores leadership and governance structures for integrating mental healthcare into Ghana’s primary healthcare (PHC) system, appears to contribute to existing knowledge, deepen our understanding of current challenges, and has the potential to inform policy and practice to strengthen mental health services.

However, I have the following concerns that needs to be addressed for better clarity and further improvement:

Title: While leadership and governance of mental healthcare are relevant across various levels, from global to local contexts, the current manuscript focuses specifically on leadership and governance related to integrating mental healthcare into primary healthcare (PHC). This specificity should be clearly reflected in the title to enhance clarity for readers and avoid any ambiguity.

To the best of my understanding, the current manuscript focuses on “leadership and governance for integrating mental healthcare at the community level”

Introduction: While the introduction seeks to emphasize the scope and importance of the problem under study and identify existing gaps, the arguments and claims should be substantiated with contemporary and specific evidence. For instance, arguments claimed from “WHO’s Integrating Mental Health in Primary Health Care Package (https://www.emro.who.int/mnh/publications/integrating-mental-health-in-primary-health-care-package.html) would provide more relevant and specific context for mental healthcare compared to those references on generic health care context.

Additionally, the authors should verify the accuracy of references cited throughout the manuscript. For example, references 10 and 13 appear to be identical and need correction.

The sentence in lines 116-117 needs to be completed to reflect the context under enquiry. “Improving and sustaining community mental health is integral to overall health system strengthening efforts ………………...”

The authors should reevaluate the use of the term “indicators” in line 119, considering the methodology employed and the subject matter explored.

Objectives: The authors should ensure coherence between the title, abstract, and stated objectives. In the aim, the phrase “among others” appears to dilute the focus, making the primary topic of “Leadership and Governance,” on which the case is built, seem secondary. Moreover, “oversight, coordination, and general stewardship of mental healthcare at the PHC” is distinct from “integration of mental healthcare at the PHC.” The authors should clarify and align these concepts for better consistency.

Additionally, the authors might consider using a more precise and technically appropriate term for “oversight” that complements better with the term “coordination.”

Materials and methods: The authors should clearly define the primary purpose of both the qualitative and quantitative approaches to enhance clarity.

In the section on the study area, the text refers to Figure 1, while the figure title labels it as Figure 2.1. Additionally, the map does not display the three ecological zones. The authors should clarify the relevance of mentioning the three ecological zones and their connection to the context of the study, if applicable.

In Line 139, do “Sub-district” level health facilities provide primary and secondary health care services?

Sample Size: The authors should provide better clarity on sample size estimation. Typically, the size of the population is not directly relevant for sample size calculation, except in cases where finite population correction is applied. Additionally, it is unclear how the sample size calculation, based on a 13% prevalence of mental health conditions, is applicable to other stakeholders such as health policy officials, healthcare managers, service providers, primary caregivers of individuals with mental health conditions, civil society organizations, self-advocates, and mental health champions.

Further, there is a lack of clarity regarding the target districts, how the population of 956,597 scales to 3,095,520 with a 13% prevalence, and how the final sample size of 1,010 was determined (the calculated sample size based on the parameters provided appears to be 183). The authors need to ensure scientific rigor and clarity by bringing more clarity in their sample size estimation process.

The authors need to clarify where and why FGDs and KIIs were used?

From Table 1, it is unclear why certain stakeholders within the entire district are excluded when it might have been more convenient to include them. In this context, the authors should clarify what criteria define "convenience" in their use of convenient sampling.

Table 1 should have footnotes (for example ??F)

The authors can provide reference for “critical descriptive ethnography approach” for the benefit of readers.

From the description, it was not clear how quantitative data was collected and how sampling was done for quantitative component. What was the language of administration of quantitative questionnaire, its transnational validity, description of response categorisation as provided in table 2a and 2b etc.,

English language and grammar require revisions at places throughout the manuscript. For example, Line 213, “However. After data collection”

In line 219, the authors should clarify whether they are referring to "key indicators" or "themes." If it is "key indicators," they need to explain how these were derived.

From the description it was not clear, whether some respondents were part of both qualitative and quantitative approach.

The authors should briefly describe how and why data triangulation was done.

Overall, the methodology seems weak, and the authors should provide more clarity and bring scientific rigor in describing their methodology to ensure the results are interpreted accurately.

Results: The description of results from line 243-246 appear incongruent to the table 2a.

The table legend does not clearly distinguish between Table 2a, 2b, 3a, 3b. The authors should provide separate and more distinct legends for each table to enhance clarity. The column title for table 2b need to be revised as per the description in the main text (?? regions)

The authors should provide results/thematic illustration of the actor mapping exercise (mentioned in method section) either in the results section or as supplemental material.

The rationale for using a generic questionnaire for healthcare providers and service users, who have different mandates, responsibilities, competencies and needs, needs to be justified. The same applies to how the results are presented.

It is appreciable that the authors have discussed the relevance of results by geographic region in the discussion section.

The authors should supplement the quantitative questionnaire, KII and FGD guide for better validation and interpretation of the results.

Results could be benefited by providing sub-headings or key themes (for example it was not clear how oversight, coordination, and general stewardship were presented).

Discussion: With lack of adequate information in methods section, it was not clear whether “leadership and governance of mental health care at the community level” was assessed or “perceptions about leadership and governance of mental health care at the community level" was assessed.

In lines 503-505, the description does not clearly explain how the mention of traditional and spiritual healers' activities indicates that the importance of mental health is being gradually and steadily recognized. The authors should provide clearer explanation on this point.

The discussion of the qualitative findings is relatively limited compared to the quantitative findings, which raises concerns about the relevance and added value of the qualitative component in relation to the quantitative component.

The lines 521 to 525 which are politically sensitive, “The Minister of Health needs to make a strong case to get the President and Cabinet aware of the urgency of addressing the nation's mental health problems through a thorough implementation of current policies and laws supported by funding to expand access to mental healthcare. A presidential initiative on mental health might help the existing plan to build 111 hospitals in Ghana, including two psychiatric facilities” should be avoided. The authors should use technical terms and avoid terms referring to persons/personnel.

At places discussion appears not to flow from the results, for example lines 531-532, “high levels” of stigma and poor attitudes and practices towards mental health and people with mental neurological and substance use illnesses……,

Implications should also describe how findings from Ghana have global mental health relevance.

Limitations: one is not sure how identifying contextual issues on the leadership and governance in mental health services across the country is a limitation.

Reviewer #3: (No Response)

7. PLOS authors have the option to publish the peer review history of their article (what does this mean? ). If published, this will include your full peer review and any attached files.

**Do you want your identity to be public for this peer review?** For information about this choice, including consent withdrawal, please see our Privacy Policy .

Reviewer #2: No

Reviewer #3: **Yes: ** Dr Pankaj Bhardwaj

---

## [Decision Letter · Decision Letter 2]

14 Mar 2025

PGPH-D-23-02157R2

Leadership and governance for integrating mental healthcare at the primary healthcare (PHC) level: a mixed methods study in Ghana

Dear Dr. YARO ,

Thank you for submitting your manuscript to PLOS Global Public Health. After careful consideration, we feel that it has merit but does not fully meet PLOS Global Public Health’s publication criteria as it currently stands. Therefore, we invite you to submit a revised version of the manuscript that addresses the points raised during the review process.

We look forward to receiving your revised manuscript.

Kind regards,

Roopa Shivashankar, MD, MSc

Academic Editor

Additional Editor Comments (if provided):

Reviewers' comments:

Reviewer's Responses to Questions

**Comments to the Author**

1. If the authors have adequately addressed your comments raised in a previous round of review and you feel that this manuscript is now acceptable for publication, you may indicate that here to bypass the “Comments to the Author” section, enter your conflict of interest statement in the “Confidential to Editor” section, and submit your "Accept" recommendation.

Reviewer #2: (No Response)

2. Does this manuscript meet PLOS Global Public Health’s publication criteria ? Is the manuscript technically sound, and do the data support the conclusions? The manuscript must describe methodologically and ethically rigorous research with conclusions that are appropriately drawn based on the data presented.

Reviewer #2: Partly

3. Has the statistical analysis been performed appropriately and rigorously?

Reviewer #2: N/A

4. Have the authors made all data underlying the findings in their manuscript fully available (please refer to the Data Availability Statement at the start of the manuscript PDF file)?

Reviewer #2: Yes

5. Is the manuscript presented in an intelligible fashion and written in standard English?

Reviewer #2: Yes

6. Review Comments to the Author

Reviewer #2: While the authors have made efforts to address the concerns, the revisions seem far from sufficiency.

1. The authors might consider using a more precise and technically appropriate term for “oversight” that complements better with the term “coordination.” I presume the oversight the authors meant pertains to “supervision” or “monitoring”

2. The figure title in the figure still label it as Figure 2.1 instead of Figure 1.

3. Regarding sampling terminology, the authors might reconsider using “stratified sampling” instead of “stratified purposive sampling.” The term “purposive” is typically associated with non-probability sampling, and its usage should be reconsidered. Since the sampling units (respondents) are defined primarily by eligibility criteria, the terminology should accurately reflect the sampling approach.

4. In the revised manuscript, the assumptions for sample size calculation have not been provided. Additionally, references 65–68 appear to be incorrectly placed. Without sufficient details, it is difficult to evaluate the veracity of the final estimate.

5. The authors should incorporate their response-10 into the footnotes of Table 1. Furthermore, in Table 1, the meaning of "F" is unclear. While it might be inferred as “Female,” the table should be self-explanatory to avoid ambiguity.

6.It is also unclear why certain paragraphs, such as “Ethics Approval and Permissions” and “Inclusivity in Global Research,” were removed while some were added without explanation. These unexplained changes complicate the review process, making it challenging to assess the revisions without starting from scratch.

7. There are quite lot of discrepancies (with additions, deletions and modifications) between the previous and revised versions particularly in the methodology section that raises serious concerns. To improve clarity, the authors should explicitly mention which paragraphs or lines were added or deleted and provide a justification for these changes.

8. Finally, the lack of a point-by-point response and the use of a generic response by combining the queries makes it difficult to determine whether the concerns have been adequately addressed for the benefit of the readers. Providing a detailed point-by-point response would enhance transparency and facilitate the review process.

7. PLOS authors have the option to publish the peer review history of their article (what does this mean? ). If published, this will include your full peer review and any attached files.

**Do you want your identity to be public for this peer review?** For information about this choice, including consent withdrawal, please see our Privacy Policy .

Reviewer #2: No

---

## [Decision Letter · Decision Letter 3]

26 Jun 2025

PGPH-D-23-02157R3

Leadership and governance for integrating mental healthcare at the primary healthcare (PHC) level: a mixed methods study in Ghana

Dear Dr. YARO,

Thank you for submitting your manuscript to PLOS Global Public Health. After careful consideration, we feel that it has merit but does not fully meet PLOS Global Public Health’s publication criteria as it currently stands. Therefore, we invite you to submit a revised version of the manuscript that addresses the points raised during the review process.

We look forward to receiving your revised manuscript.

Kind regards,

Medhin Selamu Tegegn

Academic Editor

Reviewers' comments:

Reviewer's Responses to Questions

**Comments to the Author**

1. If the authors have adequately addressed your comments raised in a previous round of review and you feel that this manuscript is now acceptable for publication, you may indicate that here to bypass the “Comments to the Author” section, enter your conflict of interest statement in the “Confidential to Editor” section, and submit your "Accept" recommendation.

Reviewer #2: (No Response)

Reviewer #4: (No Response)

2. Does this manuscript meet PLOS Global Public Health’s publication criteria ? Is the manuscript technically sound, and do the data support the conclusions? The manuscript must describe methodologically and ethically rigorous research with conclusions that are appropriately drawn based on the data presented.

Reviewer #2: Yes

Reviewer #4: Yes

3. Has the statistical analysis been performed appropriately and rigorously?

Reviewer #2: N/A

Reviewer #4: Yes

4. Have the authors made all data underlying the findings in their manuscript fully available (please refer to the Data Availability Statement at the start of the manuscript PDF file)?

Reviewer #2: Yes

Reviewer #4: Yes

5. Is the manuscript presented in an intelligible fashion and written in standard English?

Reviewer #2: Yes

Reviewer #4: Yes

6. Review Comments to the Author

Reviewer #2: The authors have addressed most of the concerns diligently. Before accepting, the authors can recheck the lines 172-174 and to confirm whether it is "0.00115403 proportion of the population" or 0.115403% of the population". With the current description of 0.115403 proportion of the population, the sample comes to 110394 instead of 1104.

Reviewer #4: Abstract

The author indicated that the study is a mixed method; hence, the author should add a key major finding from the quantitative analysis to the abstract.

Introduction

The author should rephrase the first sentence in the introduction section. The phrase “mental health conditions” appears three times, making the sentence wordy. This could be rephrased as: "Only a small proportion of people with mental health conditions receive treatment, despite effective interventions that address these conditions, prevent associated impact, reduce their prevalence, and promote overall mental wellbeing."

Line 130: add a comma after the word "approaches"

Line 176: remove the website link since "(64)" indicates this

Lines 433-436: The structure of the statement makes it difficult to see the comparison between groups. The author needs more clarity and flow with a logical structure. The phrase "by the location of the respondents was partial, 41.9% (348)" is unclear. What was the percentage representing? Could it be that the author meant that, overall, 41.9% (348) of the respondents had partial awareness of Ghana's mental health policy implementation at the PHC level? If that is the case, it will contradict the next statement, "This was followed by 35.7% (296) of respondents of the three geographical locations". The author should use a more concise and clear reporting method used in academic journal.

Line 575: Add a comma after the word "low"

Line 638: add bracket open in front 2.43

Line 674: Change the capital letter "L" in the word "Law" to a small letter l.

7. PLOS authors have the option to publish the peer review history of their article (what does this mean? ). If published, this will include your full peer review and any attached files.

**Do you want your identity to be public for this peer review?** For information about this choice, including consent withdrawal, please see our Privacy Policy .

Reviewer #2: No

Reviewer #4: **Yes: ** Abiona Taiwo Olufemi

---

## [Editor Report · Decision Letter 4]

29 Jul 2025

Leadership and governance for integrating mental healthcare at the primary healthcare (PHC) level: a mixed methods study in Ghana

PGPH-D-23-02157R4

Dear Authors,

We are pleased to inform you that your manuscript 'Leadership and governance for integrating mental healthcare at the primary healthcare (PHC) level: a mixed methods study in Ghana' has been provisionally accepted for publication in PLOS Global Public Health.

Best regards,

Medhin Selamu Tegegn

Academic Editor